# A Proposal for a Comprehensive Human–Animal Approach of Evaluation for Animal-Assisted Interventions

**DOI:** 10.3390/ijerph16224305

**Published:** 2019-11-06

**Authors:** Henrik Lerner

**Affiliations:** Department of Health Care Sciences, Ersta Sköndal Bräcke University College, 116 28 Stockholm, Sweden; henrik.lerner@esh.se

**Keywords:** one welfare, AAI, capability approach, horse, equine, animal welfare, IAHAIO white paper

## Abstract

*Background*: An important field of human–animal interactions is animal-assisted interventions (AAIs), which refers to research on human–animal interactions in order to promote or facilitate health or education in humans. Very few studies among the rich literature on AAIs seem to include aspects of animal welfare and/or animal ethics. Also, very few studies on AAIs have a comprehensive human–animal approach that studies animals, humans, and the relations between them at the same time. This paper tries to argue for and present a possible comprehensive human–animal approach to evaluate AAIs. *Methods*: A combination of the species and role approach proposed by Lerner, the capability approach proposed by Nussbaum, and a modified comprehensive human–animal approach to evaluate AAIs proposed by Lerner and Silfverberg was philosophically analyzed. *Results*: This paper shows that the combination is the modified role and species version of the capabilities approach, and by following it one could do a comprehensive human–animal approach of an evaluation of AAIs. *Conclusion*: Although the aim was reached for horses and animal-assisted therapy, further work needs to be done for all species suggested in the IAHAIO (International Association of Human–Animal Interaction Organizations) White Paper as well as for all branches of AAIs in order to establish this comprehensive human–animal approach.

## 1. Introduction

An important field of human–animal interactions is animal-assisted interventions (AAIa), which refers to research on human–animal interactions in order to promote or facilitate health or education in humans. The International Association of Human–Animal Interaction Organizations (IAHAIO) claims that three branches of AAIs exist, animal-assisted activity, animal-assisted therapy, and animal-assisted education [1].

Very few studies among the rich literature on AAIs seem to include aspects of animal welfare and/or animal ethics [2]. Also, very few studies on AAIs have a comprehensive human–animal approach that studies animals, humans, and the relations between them at the same time [3]. These two aspects combined indicate that many studies might be too narrow in their scope when evaluating AAIs. From other research areas studying human–animal interactions, there are also other interesting proposals. There have been suggestions that in a work environment where both animals and humans are present, promoting animal welfare might improve human well-being and vice versa [4]. The new approach of one welfare (kindred of the more familiar one health approach) advocates a research agenda where a comprehensive human–animal approach on welfare issues is aimed for when both humans and animals are present [5]. Altogether, this paper tries to argue for and present a possible comprehensive human–animal approach to evaluate AAIs, which could be used as a standard for evaluating good welfare for both animals and humans.

## 2. Animal Value and Animal Health

In the scientific discussions on how to define animal welfare and animal health, mental animal health is often considered as rich as in humans, at least for species that share a similar nervous system compared to humans [6,7]. In animal ethics, depending on ethical theory, the reason to give animals moral standing differs. In order to have moral standing the animal needs to share specific criteria with humans, such as the ability to suffer in utilitarianism or rationality in deontological ethics. The strongest value possible, the intrinsic value, is for animals linked to a deontological view where the common denominator is rationality.

Legislation and guidelines could also be helpful in improving a comprehensive human–animal approach. The most powerful of these is national legislation, and changes in national legislation can make binding forces to improve welfare for both humans and animals. In most legislations worldwide, animals receive far less welfare than human animals. Animals are seldom regarded as having intrinsic value. However, quite recently, in line with frontline changes in national legislation elsewhere, even the new Swedish Animal Welfare Act [8] stated that humans should respect the animal and the animal’s welfare together with promoting animal welfare in a wide sense. The preparatory work of the new Swedish legislation states that animals should be regarded as having intrinsic value, not as much as the human intrinsic value, but far more than the earlier instrumental value that animals had before [9]. This is in line with developments in animal and environmental ethics.

## 3. Species-Specific and Role Approach

Lerner [10] argued that discussions on animals often falsely lumped all animals into one group without recognizing differences due to species and the role humans give to animals. This has been the case for many works within scientific fields analyzing animal ethics, animal welfare, and legislation on animals. Modern scientific discussion needs to address these two aspects when considering animals. The species-specific aspects of an animal are rooted in its biology, while the role aspect is based on how humans value the animal and sets prerequisites in the task or area where it lives and perform. In Lerner’s theory, the role aspect divides animals into groups depending on whether they are, for example, pets, competition animals, hunting animals, or production animals. The role could sometimes be more important to define than the species, and his analysis of legislation showed that national legislation was sometimes structured depending on role aspects rather than species aspects. A proper comprehensive human–animal approach would therefore need to consider both species-specific and role aspects in improving human and animal well-being or welfare at the same time.

## 4. IAHAIO White Paper

There are no available comprehensive human–animal approach-guidelines. In the IAHAIO White Paper [1], intended as an international guideline defining AAIs as well as stating guidelines for the welfare of animals involved, the main aim is to bring attention to animals and raise awareness that animals need to be treated properly and not only as instruments for human health. The guideline rules out that only a certain number of species are well suited for AAIs and only those should be used. Only domesticated species of animals are allowed, ruling out all individuals of species of wild origin. Not all animals in a species are proper, only those with a good disposition for AAIs should be used. Both these statements clearly show that the guideline follows a species-specific (ruling out some species not suited) and role (individual that has a disposition for AAIs) approach. One could also argue that the role aspect is present in the guidelines. Here, an AAI is a part of the role of domesticated animals and cannot be a part of the role of wild animals in the wild or wild animals in human care (such as residents in zoos, enclosures, etc.).

How to monitor and promote animal health is also addressed, but mental health in animals is most probably included but not explicitly expressed in specific statements. The IAHAIO guideline would be improved if it clearly stated what definition of health in animals is used. Several definitions of health in animals exist and some of them could be used for both animal and human health. For example, the WHO definition of health in humans is also used in discussions in veterinary medicine [7]. The European project of Welfare Quality^®^ aims at achieving a monitoring scheme for animal welfare that also evaluates aspects of mental health. Still, mental health has been hard to incorporate in the scheme, especially, for example, for horses [11], but future research in animal welfare will probably solve this. The IAHAIO guideline is a minimum level guideline and one would argue for something more in a comprehensive human–animal approach so that both mental health for humans and animals could be evaluated.

## 5. The Capability Approach

Is there already a theory that could be adapted to a comprehensive human–animal approach and work as a standard for AAI that is above the minimum level proposed in the IAHAIO White Paper? The philosopher Martha Nussbaum has in her work stated that her capability approach could be adapted to both humans and animals [12,13]. Her original capability approach consists of 10 capabilities that each individual human has. In order to have a good life and preserve dignity, all of these 10 capabilities need to be present, at least at a sufficient level. Dignity is very strongly linked to intrinsic value in this theory. None of the capabilities could be exchanged with some of the other capabilities. The capabilities concern matters, such as physical health, mental health, political aspects, and also relation to other species. Nussbaum stresses that animals are also able to have these capabilities, but she adds that this needs to be analyzed with regard to dignity and how the capabilities would look like in animals.

## 6. Role and Species Capability Approach

Lerner and Silfverberg [14] developed Nussbaum’s capability approach in such a way. They also followed Lerner’s [10] species-specific and role approach focusing on horses in animal-assisted therapy, a branch of AAI. By listing each capability and analyzing each of these with regard to humans and animals in this specific setting, the analysis shows that all capabilities could be applied for both species, with the implication that the modified capabilities approach could be used within AAI as a comprehensive human–animal approach and work as a standard for evaluating welfare for both animals and humans. There are three possible levels: Harmful, non-harmful, and beneficial. At the harmful level, capabilities are reduced. As long as all capabilities present at the beginning of AAI are preserved, it reaches the non-harmful level. At the beneficial level, some capabilities are improved for the human, the horse, or for both.

## 7. Conclusions

As argued in this paper, very few studies on AAIs have a comprehensive human–animal approach. In order to find such a comprehensive approach, one should acknowledge that animals can have intrinsic value. National legislation and international guidelines provide help in reaching this; however, there is still a need for a more comprehensive approach.

Lerner and Silfverberg [14] have shown that the modified role and species version of the capabilities approach could be used as a standard for evaluating AAI. However, they have only analyzed one branch of AAI (role) and horse (species). Therefore, further work needs to be done for all species suggested in the IAHAIO White Paper as well as for all branches of AAIs [1]. The role and species version of the capabilities approach also fulfils modern national legislation, such as the Swedish Animal Welfare Act, which regulates that humans should respect the animal and the animal’s welfare, and it is in line with the claims for a comprehensive human–animal approach within the fields of AAIs and one welfare [3,5].

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
