# Peer review of "A Proposal for a Comprehensive Human–Animal Approach of Evaluation for Animal-Assisted Interventions"

_ijerph, 2019, doi:10.3390/ijerph16224305_

Round 1
Reviewer 1 Report
This is a very short paper suggesting that the welfare of animals when used in human assisted therapy is important. This is not news or original and if it is to advance the debate, then we need some standards, not just a statement that it is imporant. The authors have not done their homework thoroughly, quoting only one or two of the thinkers and writers on this subject who have rather fixed ideas that need much more thought ( eg the "capability" idea).
It may be of some use in its present form to those doctors who have never heard of animal assisted therapy but I would have thought that anyone reading this journal would have! It needs a much more thorough treatment if it is to be of any interest, how, why, where, when, should the animal ( of a given species) be given choices and how to provide a life of quality for him/her, etc etc. There are plenty of references on how to understand this better and move towards a greater understanding of another species, and individual's epistemology ( this presumably is the"role" function but this is not clear either).
It should include a proper argument for intrinstic value of the species considered, why where when etc, because doctors and careers alike may not have much idea what this involves and why. Then how to really provide a life of quality to any species without being prescriptive and outlawing some species without any discussion.
As someone at present involved with developing standards for any mammal keeping in any situation, I find this paper inadequate as it stands, but with expansion, further carefull literature review and discussion of what is meant by the various terms used, it could be helpful to ensuring that the homocentric doctor or human carer of other humans, thinks about the animals' needs at least a little!
Author Response
Thank you very much for your helpful comments in order to improve my text. Here I make my response to your comments point by point.
Standards of animal welfare - throughout the paper I have tried to clarify standards of animal welfare and the road ahead. The "role" function is not clear - I can understand that the role-aspect was somewhat confusing in my earlier version. I have now rewritten the theory behind the use of animal roles as I work with it. A proper argument for intrinsic value - I have added a part discussing intrinsic value.Using your comments and thoughts I have tried to improve the text throughout in order to show more clarity on my standpoints.
Reviewer 2 Report
Lerner reviewed about the current literature concerning the existing approaches used to evaluate the Animal Assisted Interventions, an emerging research field that includes animals in health, education and human services. Furthermore, the author in this study concept suggests applying the Martha Nussbaum’s capability approach as tool in order to evaluate the ethical situation for all animals included in IAHAIO White Paper and for all branches of Animal Assisted Intervention.
The manuscript is well written and the used bibliography is the adequate. However, in my opinion, the current version of manuscript only targets a limited number of people. Furthermore, some aspects need to be deepened and the objective of this study concept should be better defined. All suggestions are addressed below point by point:
Under INTRODUCTION:
The author should list and briefly define all branches of Animal Assisted Interventions (Animal Assisted Therapy, Animal Assisted Education, Animal Assisted Activity and Animal Assisted Coaching)Under Animal value and animal health:
The author states: “Legislation and guidelines could also be helpful in improving a comprehensive human-animal approach”. Immediately after, the author cites the Swedish Animal Welfare Act as a good reference for that. In my opinion, this paragraph adds little to the objective of the study concept because the IAHAIO White Paper (also cited later by the author) already exists as an international reference for the human-animal approach. I suggest to the author to revise this paragraph and better define its purpose.Under IAHAIO White Paper:
Please define what the IAHAIO is and which is their mission. The author state: “The guideline rules out that only a certain number of species are well suited for AAI and only those should be used”. Which are these species? Does the author suggest that the number of species should be increased? Please, clarify it in the paragraph. The author suggests that IAHAIO white paper should include specific statements about how monitoring the mental health state of animal. The author should support this citing some research aimed to improve this part of guidelines.Under CONCLUSION:
The author should better resume and include all concerns raised in each paragraph of this manuscript turning them out as future perspectives to taking into account to advance in the AAI research field.Author Response
Thank you very much for your helpful comments in order to improve my text. Here I make my response to your comments point by point.
Under introduction - I have briefly defined the branches stated in the IAHAIO White paper. Under animal value and animal health - This section is rewritten in order to meet the suggestions made by the reviewer. Under IAHAIO White Paper - I have added information on which species as well as clarified my position on the matter. I have also expanded on the monitoring on mental health in order to meet the suggestions of the reviewer. Under Conclusion - I have added sentences to cover more of the paragraphs in the text, hopefully meeting the suggestions of the reviewer. However, I prefer a rather short conclusion.Round 2
Reviewer 1 Report
The paper is improved, but as far as the debates on animal welfare and using animals for human therapy goes, I am not sure it adds much. However it is important to bring some ideas of animal welfare to any reader using animals for human therapy therefore I would think it should be published now, but it really needs a longer treatment with the main arguments reviewed properly. It is a large subject, but at least if it is aired it may draw attention to it.
The English in the added bits in yellow needs correcting, grammar is poor.
Reviewer 2 Report
The revised manuscript is much improved and all of the comments have been addressed satisfactorily.